# Entrepreneurial Interest and Entrepreneurial Competence Among Spanish Youth: An Analysis with Artificial Neural Networks

**Isabel Luis-Rico [1,\*], María-Camino Escolar-Llamazares [2,\*], Tamara De la Torre-Cruz [1], Alfredo Jiménez [3], Álvaro Herrero [4], Carmen Palmero-Cámara [1] and Alfredo Jiménez-Eguizábal [1]**

[1] Departamento de Ciencias de la Educación; Universidad de Burgos, 09001 Burgos, Spain; tdtorre@ubu.es (T.d.l.T.-C.); cpalmero@ubu.es (C.P.-C.); ajea@ubu.es (A.J-E.)

[2] Departamento de Ciencias de la Salud; Universidad de Burgos, 09001 Burgos, Spain

[3] Department of Management; KEDGE Business School, 33400 Talence, France; alfredo.jimenez@kedgebs.com

[4] Grupo de Inteligencia Computacional Aplicada (GICAP); Departamento de Ingeniería Civil, Escuela Politécnica Superior; Universidad de Burgos, 09006 Burgos, Spain; ahcosio@ubu.es

**\*** Correspondence: miluis@ubu.es (I.L.-R.); cescolar@ubu.es (M-C.E.-L.); Tel.: +34 667611863 (ext. 09001) (I.L-R) +34 619285868 (ext. 09001) (M-C.E.-L.);

**Abstract:** Studies of the socio-economic function of entrepreneurship have emphasized the critical role that entrepreneurial competence and its implementation play at different stages of the education system. In this paper, as a research objective, we seek to determine the entrepreneurial interest of Spanish youth aged between 15 and 18 years of age enrolled in formal secondary education programs, an initial stage in the development of entrepreneurship. A previously validated ad hoc questionnaire is applied through simple random sampling to 1764 students at secondary school in Spain. A descriptive cross-sectional study is carried out. The analysis is done with Artificial Neural Networks (ANNs), a technique that reduces the high dimensionality of data through Cooperative Maximum Likelihood Hebbian Learning (CMLHL), applying neurocomputational methods to the educational sciences. We find as key results that Spanish youth express a medium level of interest in entrepreneurship. Analysis with ANNs shows that education in entrepreneurial competence is an influential aspect of interest in entrepreneurship. As a conclusion, our results suggest that educational and curricular reforms must be undertaken to promote the development of entrepreneurial competence at various stages of education in order to increase interest in entrepreneurship.

**Keywords:** youth; secondary education; entrepreneurial intention; entrepreneurial interest; psychological variables

## 1. Introduction

Entrepreneurship is a key component of social and economic development, due to its potential positive impact on multiple beneficial outcomes including, among others, innovation, competitiveness, job creation, and personal success [1]. Consequently, the scientific literature is focusing more attention on its determinants and outcomes [2,3].

Entrepreneurship has stimulated interest in the economy. However, the need to attribute multidimensional meaning to entrepreneurial action, as well as the need for its consideration in public policy, explains the interest in its educational aspects [4]. The relevance of entrepreneurial competence is shown in the arrangement of competences that relate to active, participative, and transformative citizenship [5–8].

Moreover, as indicated by Escolar-Llamazares et al. [9], in the European Union, and notably in Spain, young people are among those most affected by the crisis and imbalance of the economic system and they are frequently unaware of the opportunities that entrepreneurship offers as an increasingly accessible alternative. In this sense, several investigations have focused on youth entrepreneurship [10–13]. Particularly in Spain, the percentage of potential entrepreneurs in 2016 was 6.1%, half as many as in 2012 [14]. More recent data show that, in 2018, up to 53.1% of the Spanish population between 18 and 64 years considered entrepreneurship a good professional option. In fact, this figure has remained approximately constant at the 53%–54% level since 2013, yet in the period from 2010 to 2012, it was around 65%–70%. This decline coincides with the economic crisis that Spain suffered and has possibly made people more aware of the risks and difficulties that entrepreneurs assume, leading to the perception that entrepreneurship is not such an attractive option [15].

In terms of age, one of the most distinctive features that characterize the people involved in each phase of the entrepreneurial process, according to Global Entrepreneurship Monitor (GEM) [15], is that in Spain the average age of people identified as potential entrepreneurs was 37.3 years in 2018. People between 35 and 44 years accounted for the subset of greater relative weight, both in terms of the potential entrepreneurial population (31.2%) and of the initial entrepreneurial population (30.1%). In contrast, the age group with the lowest relative weight was 55 to 65 years in the case of the potential entrepreneurial population (7.5%) and 18 to 24 years in the case of the entrepreneurial population in the initial phase (5.4%) [15].

Given the potential benefits of entrepreneurship for young people and, simultaneously, the low figures of young entrepreneurs, this relatively paradoxical situation raises the question of whether there is a preliminary process that stimulates rates of potential entrepreneurship at an earlier stage. Thus, the goal of this study is to analyze the relationship between the entrepreneurial interest of Spanish youth students aged between 15 and 18 years of age, their perceptions of the presence of aspects linked to entrepreneurship in their education, and their socio-demographic characteristics. Our investigation enlarges the study of entrepreneurial interest, as current investigations have centered mainly on young university graduates.

*Review of the Literature and Hypothesis*

Entrepreneurial activity and the psychological, socio-educational, and relational variables involved in its processes [16] have been the areas of interest in the literature in this field, in order to determine which factors influence entrepreneurial initiative [17]. One of the findings concerns the multi-dimensional nature of entrepreneurship, understood as the creation of ideas, businesses, and patents, as well as their embryonic management [18]. As the theory of planned behavior points out [19–21], it is in the process prior to action where entrepreneurial intention and interest, and their influential variables, assume relevance in explaining two inter-related processes (the discovery of opportunities and their exploitation), which leads to the subsequent entrepreneurial activity [9,22].

Before continuing, we believe it is important to clarify the concept of opportunity. Opportunities are essential for entrepreneurship [23–26], to the point that entrepreneurship has been conceptualized as the process by which people (either on their own or inside organizations) look for opportunities [27]. Already in the year 2000, Shane and Venkataraman [28] established that entrepreneurship is an opportunity-based behavior and that the entrepreneurial act focuses mainly on the questions of how, by whom, and with what effects opportunities are discovered, evaluated, and exploited to create goods and future services [27]. Shane and Venkataraman [28] defined opportunities as situations in which new goods, services, raw materials, and organization methods can be introduced and sold at a cost higher than their production cost [29].

Although opportunity is central to both research on and the practice of entrepreneurship, it is still a vague construct. Likewise, the dynamics of opportunity recognition, selection, and seizing are conceptually undefined and empirically hard to reach [27,29,30]. Clausen [29] pointed out that discussions among researchers about the nature of opportunities have possibly hindered theoretical development, testing, and empirical research. Researchers have discussed the nature of entrepreneurial opportunities, including their ontological and epistemological status [25,29,31,32]. At

this moment, the focus of the debate is whether opportunities result from exogenous market gaps or endogenous enactments [29,31,33].

As noted by Clausen [29], at least three perspectives can be identified in the opportunity debate. The first, the discovery perspective, argues that opportunities are real external entities and opportunities for profit. These profit opportunities emerge from market imperfections that individuals can discover. Individuals may perceive these profit opportunities because they are more alert, have superior cognitive skills, and/or perform an active search [29,34,35]. The second, the creation view, argues that opportunities are social constructions implemented through the entrepreneurial volition of individuals. In this perspective, opportunities begin as entrepreneurial cognitions that the entrepreneur objectifies [33]. Entrepreneurs can fail in their attempt to introduce a market offering because they lack the necessary abilities and competences or are simply unlucky. However, entrepreneurs can also succeed in the absence of a clear market by creating a demand for their offerings [23,29,36,37]. The third, the actualization perspective, conceptualizes opportunities as propensies. Opportunities are real, but they cannot be objectively measured or detected. Entrepreneurs can access them only through their imagination or beliefs, which are subjective. Thus, while opportunities are real entities, they can be known only after they have been turned into profits [29,33,38].

As Clausen [29] pointed out, there has been a heated discussion among scholars over the nature of opportunities. Arguably, there has also been some confusion, as opportunity has been seen as a single insight at different moments in time, ignoring process dynamics [33]. To move beyond discussion, it is fruitful to think about opportunity as a wide concept that integrates diverse perspectives and intellectual positions within an academic research field [29,33].

Building on the definition of the process of entrepreneurship from the Global Entrepreneurship Monitor [14,15,39], entrepreneurial intention forms part of potential entrepreneurship and reflects the notion that it is the population between 18 and 64 years of age who expresses entrepreneurial intention over the next three years.

In this sense, our study is directed towards the stages leading up to the formation of entrepreneurial intention [40]. Specifically, we are referring to the entrepreneurial interest of youth students between 15 and 18 years of age. It is in the educational system where these students must acquire knowledge, attitudes, and skills related to entrepreneurial competences [41,42]. It is known that formative variables have been shown to be directly related to entrepreneurial intention and survival, and it is precisely during the obligatory secondary education stage when they are most developed [43–45]. This situation leads us to an understanding of the level of entrepreneurial interest among the young population between 15 and 18 years of age. For this reason, we establish the objective of conducting research to determine the level of entrepreneurial interest (high, medium, low) among young students between 15 and 18 years of age.

In the same manner as that of Fernández and Reyes [46] and Martínez Rodríguez [47], in our study, we understand entrepreneurial competence as a competence that allows people to develop an entrepreneurial project through which to generate economic growth and social cohesion, being configured as an integrated social project.

This conception addresses competences from a holistic perspective, not only from an economic vision oriented towards excellent job performance. The development of entrepreneurial skills must be strengthened from an educational perspective. Skills such as creativity, personal and group initiative, problem-solving, and valuation and the assumption of economic risks, the development of business plans and projects, adequate solutions, etc. must be worked on [46,47]. In fact, multiple studies underscore the importance of education in affecting attitudes towards entrepreneurship [46,48], with a particularly remarkable effect in the young population. For example, Kurowska-Pysz [10] analyzed the effectiveness of Academic Incubators of Entrepreneurship (AIE) in Poland with a group of young people in secondary education and higher education. The aim was to promote managerial competence in students and to formulate recommendations regarding the development of managerial skills or future entrepreneurs. Kurowska-Pysz found that students who participated in the incubator perceived the development of desirable traits and the strengthening of specific

entrepreneurial and management competences, which increased their motivation to start a business after leaving the incubator.

Similarly, Johansen [11] performed a quantitative study of former participants in Junior Achievement-Young Enterprise (JA-YE) Europe programs and found that, usually, participants in entrepreneurship education programs are more likely to become entrepreneurs as compared to students who did not participate in the programs. The study of Finisterra do Paço, Ferreira, Raposo, Rodrigues, and Dinis [12] showed a clear impact of general education on entrepreneurship and entrepreneurial activity. This has led to increased interest, among researchers, in entrepreneurship education programs. Furthermore, some works suggest that early formal entrepreneurship education can have an impact on the attitudes of students, influencing them in the direction of their future careers and thereby boosting their propensity for entrepreneurship when they become adults [13].

Previous literature have focused on determining the skills, knowledge, and attitudes that form the structure of entrepreneurial competence. They have done this by establishing the influential variables of such activities [49,50]. Nevertheless, scholars agree that there is a clear transversal and interdisciplinary component. According to the classification of competences of the Harvard Graduate School of Education [51], entrepreneurial competence is related to interpersonal competences (theory classes, practical exercises, case studies, project design, teamwork, problem-solving, presentation, and defense of assignments), intrapersonal competences (leadership, commitment and motivation, creativity and innovation, conflict and crisis management considered as tolerance of pressure, communicative capability, capacity to negotiate and make decisions, time management for personal and team work, and the capacity to search for resources), and cognitive competences (business planning, marketing, languages, computing skills, organizational planning, obtaining resources, legal aspects of business creation, and administrative management). In other words, entrepreneurial competence covers the development of business mentality, constituted by proactivity, creativity, innovation, and risk assumption, as well as a capability for project planning and management to set objectives [52,53].

If we consider that Spanish students between 15 and 18 years of age were educated in an educational system in which entrepreneurial competence was present, as stipulated in the legal framework that regulates it, their entrepreneurial interest may be related to entrepreneurial competence. This leads us to formulate the following hypothesis:

**H1:** The perceptions of young students towards their education in cognitive, interpersonal, and intrapersonal competences related to entrepreneurship throughout the educational stage are related to their entrepreneurial interest.

In terms of the variables that affect entrepreneurial interest, socio-educational factors (a social space and a workplace) have been considered relevant in analyses of entrepreneurial intention [17,22,40,54]. On the one hand, these factors are linked to sociological demographic aspects such as age [55,56], gender [57,58], level of studies, employment situation, income, civil status, and professional status [54]. On the other hand, they are also linked to empowerment in society through education and are directly related to entrepreneurship rates, thereby calling for the inclusion of entrepreneurial competence in educational curricula [9,59].

Thus, a relationship may be expected to arise between some of the socio-demographic variables and the entrepreneurial interests of young people. In this respect, we formulate the second hypothesis:

**H2:** Socio-demographic aspects such as gender, educational level, and employment of parents or tutors of Spanish youth are related to the youth's entrepreneurial interests.

## 2. Materials and Methods

### 2.1. Design

This investigation was a questionnaire-based transversal descriptive study of a population with a probabilistic sample that formed part of a project coordinated at the national level and was developed by the universities of Barcelona, Burgos, Deusto, La Rioja, and Santiago de Compostela and the National University of Distance Education. In addition, the University Pablo de Olavide de Seville and the University of Valencia assisted with data collection activities.

### 2.2. Participants and Procedure

The study population was composed of 15- to 18-year-old secondary education students enrolled in secondary school in Spain. The sampling size, 1764 students, was calculated at a confidence level of 95% and an error margin of 2.3% based on data contributed by the Ministry of Education, Culture, and Sports for the 2010–2011 academic year (Table 1).

**Table 1.** Population and sample.

|          | Population | N    | Error margin | Confidence level |
|----------|-----------|------|--------------|------------------|
| Students | 1,047,331 | 1764 | 2.3%         | 95%              |

Simple random sampling was used, and the set proportions were conserved in each of the Autonomous Regions and at each of the general tiers of education (67% of students from pre-university (high school) studies; 32.7% of students from intermediate vocational training cycles; and 10.3% of students from basic vocational training). The last sampling units were chosen during the 2013–-14 academic year through the random selection of educational centers in each Autonomous Region on the basis of two criteria: the selection of a rural center in each Autonomous Region and a proportion of one private state-assisted center for every three public educational centers. At each of the selected centers, the questionnaire was administered in a single session during the months of March and June 2014. By collecting data in 2014, we were able to use more comparable data, as all participants were enrolled at school following the same curriculum (which was modified the following year by a new educational law in Spain, though with no effect on our data). Before the application of the instruments, permission was requested from both the General Directorate of Education of each Autonomous Region and the directors of the educational centers, once they were informed about the rationale behind the research. Two previously trained researchers assisted in person at each center to administer the questionnaire, thereby ensuring a duly standardized protocol of action. Women comprised 50.1% of the sample (n = 885) while men comprised 49.9% (n = 879). The average age was 17.6 years (SD = 1.60) and 89.6% of the sample were of Spanish nationality (n = 1581).

The present study was funded by the Ministry of Economy, Industry, and Competitiveness and the European Regional Development Fund (Spain), which requires no ethics committee approvals for studies in the field of social and legal sciences.

### 2.3. Measures

The questionnaire (S1 and S2 Files) was divided into various thematic sections, aligning with the different interests of each of the collaborating universities: students; life at the educational center; leisure time; family life; health and quality of life; studies and the employment market in the future; and entrepreneurship.

A pilot test in eight Autonomous Regions was completed to validate the questionnaire, establishing the stratification of the final sample and its proportionality as criteria. The number of questionnaires amounted to 10% of the subsequent sample. The pilot survey and results were then evaluated by 14 experts from seven Spanish universities, who approved the definitive version and rated it as highly reliable. Likewise, the reliability of this questionnaire was contrasted in previous studies published by other authors [9,60,61].

This paper reports the results of the questionnaire administered to students corresponding to study blocks, the employment market in the future, and entrepreneurship (Table 2).

**Table 2.** Selection of questions from the student questionnaire.

| Num. Question (Q.) | Block | Information | Measure |
|---|---|---|---|
| Q.1 to Q.10 | Student history | Sex; Year of birth; People living with you; Occupants of the house; Family situation; Family relation; Country of origin, father, mother and student; Post code; Available money; Level of studies and professional situation of father and mother. | Different forms of designing the data in accordance with the question. |
| Q.37a. | Studies | Aspects with which studying helps. | 1 to 5 Likert Scale. |
| Q.37b. | Labor market | Aspects with which working helps. | 1 to 5 Likert Scale. |
| Q.40 | Entrepreneurship | Degree of interest in entrepreneurship (IE). | 1 to 5 Likert Scale. |
| Q.44a. | Entrepreneurship | Presence in their training activities: Theory classes; Practical exercises; Case studies; Design projects; Teamwork; Problem-solving; Presentation of work. | 1 to 5 Likert Scale. |
| Q.44b. | Entrepreneurship | Presence in their training activities: Leadership; Commitment and motivation; Creativity and Innovation; Conflict management (Tolerance to pressure); Communicative capacity; Capacity for negotiation and decision-making; Management of time to do own work and team work; Capacity to conduct a search for resources. | 1 to 5 Likert Scale. |
| Q.44c. | Entrepreneurship | Presence in their training activities: Business planning; Marketing plan; Languages; Computers; Planning organizations; Obtaining resources; Legal aspects; Business creation; Administrative management. | 1 to 5 Likert Scale. |

*2.4. Data Analysis*

Having selected the questions for inclusion in the study, we used a code to identify each data item. This resulted in a data set with 1633 items and 55 characteristics for each one, responsible for the aforesaid high degree of dimensionality of the data set.

Artificial Neural Networks (ANNs) were used due to the data properties and the need to find a vector characteristic of a pattern that would meet the investigative objective. The ANNs grouped the data and determined the orientation and characteristics of the groupings. Thus, for the data analysis, we selected the unsupervised neural architecture model of Cooperative Maximum Likelihood Hebbian Learning (CMLHL), characterized by its ability to conserve a degree of global order in the overall data set. This technique overlaid the topological order of the different neurons, following a neighborhood or a similarity approach, in which the Exploratory Projection Pursuit (EPP) technique could be applied.

As an exploratory technique, it can be applied when there is no target data to be reproduced by the neural network (as opposed to supervised learning). Hence, it fits the problem that the present paper addresses, in which there is a data set that must be analyzed and whose internal structure has previously been unknown. To do so, different alternatives can be used. CMLHL is one alternative that has previously been applied to a wide variety of problems. Because it takes higher-order statistics into account, its projections are more sparse and informative than those obtained by more traditional methods such as Principal Component Analysis [62–64].

EPP [65] is employed to resolve problems in which it is difficult to identify the internal structure of high dimensional data sets. To do so, an index must first be defined that measures the interest of a vector projection. Afterwards, the data are transformed by maximizing that index in such a way that the interest attributed to it is maximized. Diaconis and Freedman [66] affirmed that a typical random projection is Gaussian and, statistically, the most interesting projections are those at a distance from that distribution.

A neural implementation of EPP is Maximum Likelihood Hebbian Learning (MLHL) [67]. The operations of this neural model are defined as:

Feedforward step:

$$y_i = \sum_{j=1}^{N} W_{ij} x_j, \forall i$$

II.3.A

Feedback step:

$$e_j = x_j - \sum_{i=1}^{M} W_{ij} y_i$$

II.3.B

Updating of weights:

$$\Delta W_{ij} = \eta y_i \, sign \, (e_j) |e_j|^{p-1}$$

II.3.C

The inclusion of lateral connections to MLHL was proposed [68]. This yields the Gaussian Rectified Distribution (GRD). It is based on cooperative distributions that generate the model known as CMLHL. GRD is a modification of the standard Gaussian Distribution. The starting point for obtaining it is the standard GRD:

$$p(\mathbf{y}) = Z^{-1} e^{-\beta E(\mathbf{y})},$$

II.3.D

where the parameter, $\beta = 1/T$, is the inverse of the temperature. $\beta$ produces a decrease in temperature, concentrating the distribution to the minimum of the energy function. The $Z$ factor normalizes the integral of $p(\mathbf{y})$ to the unit, while $E(\mathbf{y})$ is the cost function or associated energy, expressed as:

$$E(\mathbf{y}) = \frac{1}{2} \mathbf{y}^T \mathbf{A} \mathbf{y} - b^T \mathbf{y}$$

II.3.E

The quadratic energy function, $E(\mathbf{y})$, was defined by the parameter "bias", $b$, and the symmetric matrix, $\mathbf{A}$. Depending on that matrix, $E(\mathbf{y})$ may display different curves. It should be taken into account that not all of the energy functions may be used in the GRD. The only functions that are included are those that mean that matrix $\mathbf{A}$ will comply with the property of co-positiveness, through which the directions in which the energy tends towards negative infinity are blocked, mathematically expressed as:

$$\mathbf{y}^T \mathbf{A} > 0 \quad \text{for all} \quad \mathbf{y} > 0$$

II.3.F

Cooperative Distribution in the case of $N$ variables is defined as:

$$A_{ij} = \delta_{ij} + \frac{1}{N} - \frac{4}{N}\cos\left(\frac{2\pi}{N}(i-j)\right)$$

     II.3.G

$$b_i = 1$$

     II.3.H

where $\delta_{ij}$ is Kronecker's Delta and *i* and *j* identify the input and output neurons, respectively.

As an approximation, the matrix may be simplified, as indicated in Equation (II.3.J), to accelerate the learning process, where matrix $\mathbf{A}$ is employed to modify the response of the data based on the distance between the neurons of the output layer of this neural model, in accordance with the following expression:

$$A_{ij} = (\delta_{ij} - \cos(2\pi(i-j)/N)$$

     II.3.J

The simplest algorithm is the projection of the rectified gradient method, which refers to the gradient followed by the next rectification:

$$y_i(t+1) = [y_i(t) + \tau(b - Ay)]^+$$

     II.3.K

where $[\ ]^+$ is the necessary rectification so that the values, $y$ , remain in the positive quadrant and where $\tau$ is the strength of the lateral connections between the neurons of the output layer. An appropriate value for $\tau$ must be chosen so that the algorithm converges to a stationary point (generally local minima) for the energy function [69].

Through the addition of the lateral connections, the resulting neuron model (CMLHL) is capable of identifying a certain type of global order or structure in the data set. The steps are defined in the following manner:

Feedforward step:

$$y_i = \sum_{j=1}^{N} W_{ij}x_j, \quad \forall i$$

     II.3.L

Feedback step:

$$e_j = x_j - \sum_{i=1}^{M} W_{ij}y_i, \quad \forall j$$

     II.3.M

Lateral activation:

$$y_i(t+1) = [y_i(t) + \tau(b - Ay)]^+$$

     II.3.N

Updating of weights:

$$\Delta W_{ij} = \eta y_j \, sign\,(e_j)|e_j|^{p-1}$$

     II.3.O

## 3. Results

The topological ordering of the different neurons by their neighborhood functions and similarity is shown in Figure 1. As the objective of our investigation was centered on determining the entrepreneurial interest of youth students aged between 15 and 18 years, we activated the localization

of the variable on entrepreneurial interest (P.40) with the five characteristics that provide its dimensions.

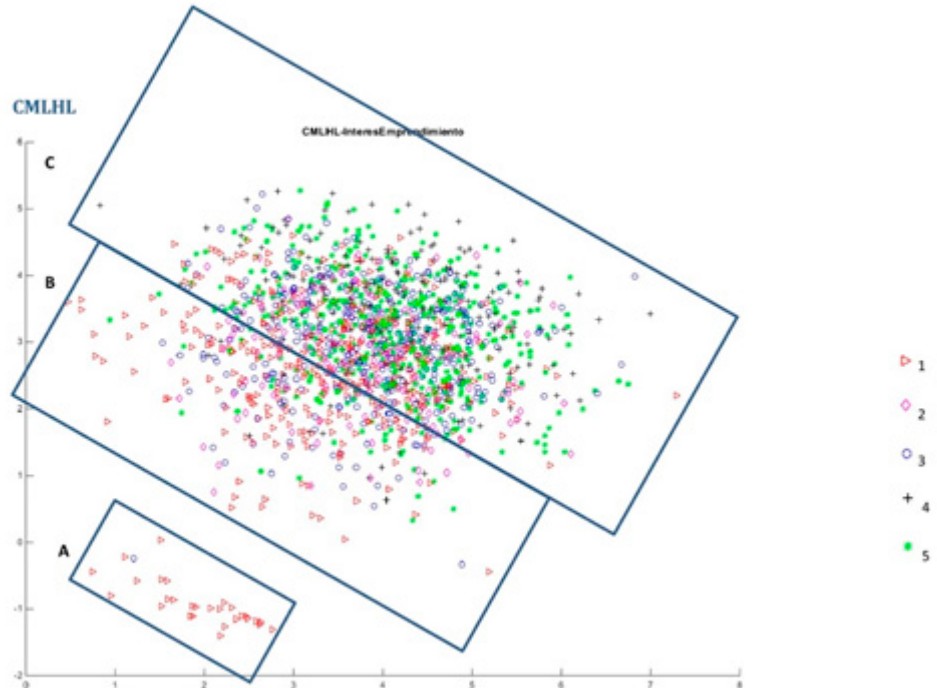

**Figure 1.** Topographic representation of the RNA of the sample with the activation of 0.40 on entrepreneurial interest. CMLHL = Cooperative Maximum Likelihood Hebbian Learning; 1 = no entrepreneurial interest; 2 = little entrepreneurial interest; 3 = some entrepreneurial interest; 4 = quite entrepreneurial interest; 5 = much entrepreneurial interest

In the resultant neural model (Figure 1), an ascendant spatial orientation can be observed as a function of entrepreneurial interest, as well as a greater proximity of neurons, identifying a structure in the data set. This is one of the characteristics of the CMLHL technique that was used. The two selected elements of density and situation indicated that the subjects that were closest to each other shared greater similarity.

Considering these two aspects of situation and density, we selected three groupings (A, B, C), permitting the extraction of the characteristics of the groupings of neurons that contribute to the confirmation of the hypotheses proposed. Thus, in Table 3, we can see the description of the characteristics of these groupings.

**Table 3.** Characteristics of the neural groups for each question selected.

| Question (Q.) | | | A | B | C |
|---|---|---|---|---|---|
| **Q.1 to Q.10 Student data** | Sex | Woman | 24% | 52% | 51% |
| | | Man | 76% | 48% | 49% |
| | Level of studies—father | None | 7% | 5% | 4% |
| | | Primary | 14% | 28% | 29% |
| | | Secondary | 58% | 37% | 38% |

| | | | | | |
|---|---|---|---|---|---|
| | | Superior | 21% | 20% | 29% |
| | Level of studies—mother | None | 10% | 4% | 3% |
| | | Primary | 31% | 36% | 28% |
| | | Secondary | 31% | 39% | 36% |
| | | Superior | 28% | 21% | 33% |
| | Professional situation—father | Employed | 55% | 59% | 59% |
| | | Self-employed | 31% | 27% | 26% |
| | | Home | 0% | 0% | 0% |
| | | Unemployed | 10% | 10% | 10% |
| | | Retirement | 0% | 2% | 3% |
| | | Pensioner | 4% | 2% | 2% |
| | Professional situation—mother | Employed | 59% | 45% | 55% |
| | | Self-employed | 14% | 16% | 13% |
| | | Home | 10% | 26% | 21% |
| | | Unemployed | 4% | 10% | 9% |
| | | Retirement | 3% | 1% | 1% |
| | | Pensioner | 10% | 2% | 1% |
| **Q.37a Studies** | Aspects with which studying helps | Having success in life | 1 | 3.4 | 4.3 |
| | | Finding work | 1 | 3.6 | 4.4 |
| | | Relating to colleagues | 1.1 | 3.2 | 4 |
| | | Earning money | 1 | 3.2 | 4 |
| | | I prefer looking for work over continuing to study | 3.1 | 2.5 | 2 |
| **Q.37b Employment market** | Aspects with which working helps | Becoming more independent | 1 | 3.4 | 4.2 |
| | | Collaborating with the family | 1 | 3.2 | 4 |
| | | Feeling useful | 1 | 3.4 | 4.3 |
| | | Having money available | 1 | 3.9 | 4.5 |
| | | Worth looking for work | 1 | 3.4 | 4 |
| **Q.40 Entrepreneurship** | Entrepreneurial interest | Degree of personal interest in entrepreneurship | 1 | 2.1 | 3.2 |
| **Q.44a Interpersonal entrepreneurial competence** | Presence in training | Theoretical activities | 1 | 3 | 4 |
| | | Practical exercises | 1 | 3.1 | 3.8 |
| | | Case studies | 1 | 2.6 | 3 |
| | | Project designs | 1 | 2.6 | 3 |
| | | Teamwork | 1.1 | 3.1 | 3.5 |
| | | Problem-solving | 1 | 3 | 3.6 |
| | | Presentation of works | 1 | 2.8 | 3.4 |
| **Q.44b Intrapersonal entrepreneurial competence** | Presence in education | Leadership | 1.1 | 2.4 | 3.2 |
| | | Commitment and motivation | 1.1 | 2.9 | 3.7 |
| | | Creativity/innovation | 1.1 | 2.8 | 3.5 |
| | | Conflict management (tolerance pressure) | 1 | 2.5 | 3.2 |
| | | Communicative capacity | 1 | 3 | 3.7 |

| | | | | | |
|---|---|---|---|---|---|
| | | Capacity for negotiation and decision-making | 1 | 2.7 | 3.5 |
| | | Management of time for own and group work | 1 | 2.6 | 3.5 |
| | | Capacity for searching for resources | 1 | 2.5 | 3.3 |
| **Q.44c Cognitive entrepreneuri al competence** | Presence in training | Business planning | 1 | 2.3 | 2.6 |
| | | Marketing | 1.1 | 2.3 | 2.6 |
| | | Languages | 1 | 2.6 | 3.6 |
| | | Information technology | 1.1 | 2.8 | 3.4 |
| | | Planning organizations | 1 | 2.6 | 3 |
| | | Obtaining resources | 1 | 2.6 | 3 |
| | | Legal aspects of firm creation | 1 | 2.3 | 2.6 |
| | | Administrative management | 1 | 2.4 | 2.6 |

As anticipated in the topological representation of the data, there was a variation in ascendant entrepreneurial interest. Thus, the grouping of A, the furthest away, showed null entrepreneurial interest (1 out of 5), while grouping B had a low entrepreneurial interest (2.1 out of 5), and grouping C showed a medium interest (3.2 out of 5). In grouping C, there was greater density and proximity between the neurons, which suggests the affinity of the projection.

If we consider the description of the groupings as a function of the different areas under study, we can see that the majority of grouping A was masculine, with a composition of 76% men, while the members of groupings B and C were balanced between men and women. With regard to the level of studies of parents, in grouping A, 58% had attained secondary-level studies, while in groupings B and C, the percentages for primary, secondary, and superior were more balanced. No notable differences were evident in the level of studies of the mothers between the groupings, although the higher percentage reached in the section "no studies" corresponded to grouping A.

We found no differences in the professional situations of both parents with regard to the percentage distributions in the groupings. The majority were salaried workers. Nevertheless, we can point to a higher percentage of pensioners in the case of mothers in grouping A, amounting to 10%, as well as higher percentages of mothers exclusively performing domestic roles in groupings B and C.

In question Q.37a, with regard to the aspects with which their studies helped, we found differences between the results for grouping A and the results for groupings B and C. Thus, the members of grouping A believed that their studies were of "no help" to them (1 out of 5) in terms of achieving success in life, nor for finding work, nor for relating to friends or earning money. Meanwhile, the replies of those in grouping B suggested that their studies helped them "somewhat" (3 out of 5), while the ratings of grouping C indicated "quite a lot" (4 out of 5). On that same question, the ratings were inverted with regard to the preference of working in the area of one's studies, with the highest values shown in grouping A (3.1 out of 5), followed by B (2.5 out of 5) and C (2 out of 5).

With regard to question Q.37b, on the aspects with which working helps (becoming independent, helping family, feeling useful, having money, and the worthwhileness of looking for work), the groupings showed differences in ratings. The ratings of grouping A were focused around values referring to "not at all", while the ratings of grouping B were focused around "somewhat", and the ratings of grouping C were focused around "quite a lot".

In question Q.44a, on the presence in their education of different activities related to the development of interpersonal entrepreneurial competence (theoretical activities, practical exercises, case studies, design projects, teamwork, problem-solving, and the presentation and defense of assignments), the results were as follows. Members of grouping A said "not at all", while in grouping B the replies were more in the region of "little" and "somewhat". Grouping C indicated "somewhat". The only aspect to be rated "quite a lot" referred to the theory classes. The aspects that received the lowest ratings were those referring to case studies and project design.

Among the ratings given in reply to question Q.44b, on the presence in their education of aspects related to entrepreneurial competence in the intrapersonal field, grouping A indicated that it had no

presence (1 out of 5), while grouping B was set around ratings of "little" and grouping C was set around ratings of "somewhat". The aspects that gained higher scores were commitment and motivation and communicative capacity, while those rated the lowest were leadership and conflict management (tolerance of pressure).

The competences referred to in question Q.44c were those with the lowest ratings across the board. In this question, the presence of cognitive entrepreneurial competences was noted in their education, with a distribution of ratings by groupings similar to the earlier one. Grouping A was situated at levels of 1, grouping B around 2, and grouping C over 2 and 3. The highest ratings were found in languages and computing, while aspects such as administrative management, the legal aspects of business creation, business planning, and marketing never exceeded ratings of 3 in any grouping.

## 4. Discussion

Regarding our research objective, the results showed a relatively low level of entrepreneurial interest of 2.9 out of 5 for students between 15 and 18 years of age. This point was due to a certain degree of inefficiency in the design of the framework for competences implemented in the secondary level of education. As Bernal and Cárdenas [41] pointed out, a paper on reforming the framework of competences in the obligatory stages of education, which leads to the acquisition of entrepreneurial competence impacts and an increase in entrepreneurial interest, has yet to appear [42]. In this sense, entrepreneurial training generates a positive effect on levels of desire felt towards entrepreneurship [59,70] because it stimulates the development of entrepreneurial behavior, thereby increasing knowledge of business creation and management. In the same way, entrepreneurial training encourages the formation of personal characteristics associated with entrepreneurship and the management of firms, thereby making students aware of the viability of self-employment as a professional path [71].

Considering H1, and in view of the data, we saw that in the resulting groupings A, B, and C, differences could be established according to the entrepreneurial interests of the students. This, in turn, allowed us to describe these students according to the entrepreneurship competences present during their training. In this way, the grouping with the highest entrepreneurial interest, C, was the one that gave the highest ratings to the presence of competential aspects related to entrepreneurship, such as leadership, commitment, motivation, and project design. Grouping C was also the grouping with higher scores in terms of considering studying to be a fundamental aspect of personal development (P37a and b). Thus, our results showed that young students' perceptions of education with respect to cognitive, interpersonal, and intrapersonal competences related to entrepreneurship throughout the secondary educational stage were, indeed, related to entrepreneurial interest. As such, the relevance of training in aspects related to entrepreneurial competence during educational development was highlighted [42,49,53].

Our results aligned with those obtained in other countries, such as those from the study of Kaya et al. [72] with German university students. These authors found that both the entrepreneurship support culture and the teaching of skills related to self-management increase the odds of future entrepreneurial activity, which reinforces the need to teach the entrepreneurial competencies in an academic environment [72]. Similarly, other studies with European and Turkish students found that entrepreneurial education is a relevant factor for providing young students with the necessary skills to act [73,74].

In relation to H2, drawing on previous literature we expected that socio-demographic aspects such as gender, age, and the level of studies of parents to have some influence on entrepreneurial activity [21,59]. However, after having analyzed the data in our study, we found no great variation between the characteristics of the groupings with respect to socio-demographic aspects. Thus, we found no support for H2. In other words, the studied socio-demographic aspects of Spanish youth students showed no relation to entrepreneurial interest. Our results differed from studies such as that of Kaya et al., who did find differences in the intention of becoming entrepreneurial depending on gender. Specifically, male students were more likely to establish their own businesses than female

students. Thus, our analysis uncovered one specific characteristic from Spanish students compared to other samples.

Nevertheless, in our study, it was significant that grouping A, which had the lowest level of entrepreneurial interest, was masculine, while there was a gender balance in the other two groupings that had percentages of entrepreneurial interest. This result is particularly interesting, as previous studies have shown that there are lower percentages of women in the phases of entrepreneurial intention [57,58]. In fact, some studies point to higher entrepreneurial aspirations among male students at the university stage [39,75]. Our results, therefore, demonstrated that the difference in gender appear at later stages after secondary education, a stage in which men and women show similar entrepreneurial interest.

Finally, as a further contribution of this research, we believe that it is important to note the innovative nature of the use of Artificial Neural Networks (ANNs) to reduce the dimensionality of the data, thereby generating patterns of study for decision-making through Cooperative Maximum Likelihood Hebbian Learning (CMLHL). This allowed us to observe the topographical representation of the neurons—a distribution linked to entrepreneurial interest, which is of use for situational analysis and to justify the use of predictive statistics and neurocomputation in the social sciences.

## 5. Conclusions

Through the analysis of groupings, we have verified the scope and the degree of support for our objective and hypotheses, from which pedagogic implications can be drawn. The results show that the entrepreneurial interest of Spanish students is medium. This result contrasts with the Spanish legislation that assigns compliance with the most ambitious of educational levels to post-obligatory secondary education, which are not compensated by the entrepreneurial interest generated. As such, there are important implications for the bodies responsible for educational policy, as well as other relevant stakeholders such as high schools, universities, incubators, commerce chambers, clusters, and networks.

At the educational policy level, our results clearly point to the need to implement educational improvements directed towards fulfilling the objectives of an education in entrepreneurship. The uncertainties, the processes of virtualization, digital transfers, and the challenges of a globalized world determine a new education endowed with entrepreneurial competence. However, that world will not be possible without a modification in the entrepreneurial interest of young people, as a direct antecedent of entrepreneurial action.

The change must start in the classroom through the inclusion of tacit and explicit knowledge of entrepreneurship. As such, not only are the bodies responsible for educational policy concerned, but high schools, universities, and other institutions related to training also need to align and coordinate. Thus, an effective transmission of knowledge, skills, and attitudes related to entrepreneurial competence, as a transversal competence, is needed. The uncertain definition of this competence within the school system complicates its development and implementation, as a closer look at the belief processes of the teaching staff towards the knowledge, capabilities, and attitudes that compose it. Hence, the competence and the development of the entrepreneurial spirit has to assume key and transversal competences, implanting this knowledge in future teachers, as the responsibility for developing entrepreneurial programs in the education system will be in their hands [76]. In this sense, active methodologies such as project-based learning (PBL) strategies based on cooperative learning and service-based learning (ApS), which pursue the development of the entrepreneurial spirit centered on social and ethical zones, can play a critical role.

Finally, in order to increase entrepreneurial interest, key stakeholders related to the practical side of the entrepreneurial environment, such as incubators, commerce chambers, clusters, and networks, need to take a more prominent and active role in promoting the various possibilities and advantages available to entrepreneurs, while at the same time providing support and encouragement to face the challenges related to it. While entrepreneurship is a critical topic that needs to be addressed at the educational stage, it is important that it does not remain an abstract concept but, rather, its

practical side is also shown by those practitioner stakeholders who can also contribute to increase entrepreneurial intention.

Our study also makes a relevant contribution by showing that at the secondary education stage we do not find a significant difference in entrepreneurial interest between men and women. Yet, there is plenty of empirical evidence of significant differences at later stages. Consequently, an important implication of our results is that all stakeholders related to the fostering of entrepreneurship need to address the reasons why these differences appear and implement urgent compensatory measures to prevent or minimize them.

The limitations of this work are related to the methodology used to restrict the generalization of the results to other contexts. A future line of investigation has been proposed to develop these types of studies among populations in other geographical areas. Further, multiple types of entrepreneurship exist, and not all of them are conducive to innovation and growth [77]. Another limitation of this work lies in the fact that we are unable to distinguish the specific type of entrepreneurship that participants were thinking about at the time they answered the questionnaire.

Finally, we have contributed to the literature on methodology in the study of education for entrepreneurship by using ANNs to reduce the dimensionality of the data, as this generates patterns of study as a decision-making base—in our case, through CMLHL. It enables the observation of a distribution in the topographical representation of the function of entrepreneurial interest, which serves as the basis for the analysis of the situation, creating a role for the use of predictive statistics and neurocomputation in the field of social sciences.

**Supplementary Materials:** S1 File. Questionnaire for students. English translation; S2 File. Questionnaire for students. Spanish Original.

**Author Contributions:** Conceptualization, I.L-R. and M-C.E-LL.; methodology, I.L-R. and M-C.E-LL.; software, A.H.; validation, C.P-C., A.J-E., and T.d.l.T-C.; formal analysis, A.H.; investigation, I.L-R., M-C.E-LL., and T.d.l.T-C.; resources, A.J-E. and C.P-C.; data curation, I.L-R.; writing—original draft preparation, I.L-R. and M-C.E-LL.; writing—review and editing, A.J. and T.d.l.T-C.; visualization, I.L-R., M-C.E-LL., A.H., and A.J.; supervision, A.J. and T.d.l.T-C.; project administration, A.J-E.; funding acquisition, C.P-C.

**Funding:** This research was funded by MINISTRY OF ECONOMY AND COMPETITIVENESS. GOVERNMENT OF SPAIN, grant number EDU 2012-39080-C07-00 a 07.

**Acknowledgments:** We would like to acknowledge all Spanish secondary education students who participated in the investigation by completing the questionnaire.

**Conflicts of Interest:** The authors declare no conflict of interest.

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
