# Peer review of "Entrepreneurial Interest and Entrepreneurial Competence Among Spanish Youth: An Analysis with Artificial Neural Networks"

_sustainability, doi:10.3390/su12041351_

Round 1

Reviewer 1 Report

The study examines the relationship between the Spanish students entrepreneurial interest, their perceptions of the presence of aspects linked to entrepreneurship in their education and their sociodemographic characteristics. This issue is rather interesting for readers, although the research are limited and they cover only  one country. There is no comparison with the similar research results in other countries. If authors extend the discussion in this way, than the paper would be more interesting for readers probably and give a better view for the conclusions from the research. However, taking into account the big changes in the European labor market as well as new trends in young people behavior this research problem is actual and should be investigated deeply.

Specific comments:

The paper structure is rather correct, but in abstract structure should be improved. This is not a section for “introduction”, the authors should shortly describe there the research goals, methods and key results.

In the section Introduction the authors should consider to add some statistical data regarding the youth in Spain, especially youth entrepreneurship (there are some information in lines 122- 125 but there is less than it should be to understand better the situation in Spain).

On the other hand the theoretical issues such as discussion of different scientific approaches, which are now included in the section: Introduction should be included in the section: Review of the literature and hypothesis authors should add some thesis regarding exactly the youth entrepreneurship. 

In the section: Materials and Methods the authors should explain why do they present the research results from 2013-2014 and why they chosen exactly this research method to verify the hypothesis. 

In the section: Conclusions the authors should also present their recommendations for the bodies responsible for the educational policy as well as for some other entities responsible for the young entrepreneurs support such as: universities, incubators, commerce chambers, clusters and networks.

Generally the issue is interesting but needs to be corrected at least  according to the above remarks.  

Good luck!

Author Response

Reviewer 1

1.) The study examines the relationship between the Spanish students entrepreneurial interest, their perceptions of the presence of aspects linked to entrepreneurship in their education and their sociodemographic characteristics. This issue is rather interesting for readers, although the research are limited and they cover only one country. There is no comparison with the similar research results in other countries. If authors extend the discussion in this way, than the paper would be more interesting for readers probably and give a better view for the conclusions from the research. However, taking into account the big changes in the European labor market as well as new trends in young people behavior this research problem is actual and should be investigated deeply.

RESPONSE: Thank you very much for your encouragement and constructive comments. We are glad that you find our paper interesting for readers. For your convenience, we include our response to each comment in italics. Following your kind suggestion, we have introduced in the discussion section studies carried out in other countries. Specifically, these studies have been added:

Minola T, Criaco G, Cassia L. Are youth really different? New beliefs for old practices in entrepreneurship. Int J Entrep Innov Manag. 2014;18(2–3):233–59. Dvouletý O, Mühlböck M, Warmuth J, Kittel B. ‘Scarred’ young entrepreneurs. Exploring young adults’ transition from former unemployment to self-employment. J Youth Stud. 2018;21(9):1159–81. Kaya T, Erkut B, Thierbach N. Entrepreneurial intentions of business and economics students in Germany and Cyprus: A cross-cultural comparison. Sustain. 2019;11(5).

Specific comments:

2.) The paper structure is rather correct, but in abstract structure should be improved. This is not a section for “introduction”, the authors should shortly describe there the research goals, methods and key results.

RESPONSE: Thank you for this remark. We have corrected the abstract structure to clearly describe the research goals, methods, and key results. 

3.) In the section Introduction the authors should consider to add some statistical data regarding the youth in Spain, especially youth entrepreneurship (there are some information in lines 122- 125 but there is less than it should be to understand better the situation in Spain).

RESPONSE: We appreciate this comment and agree with you that some statistical data could improve the introduction. Following your suggestion, we have added some statistical data on entrepreneurship in Spain and especially the situation of youth entrepreneurship.

4.) On the other hand the theoretical issues such as discussion of different scientific approaches, which are now included in the section: Introduction should be included in the section: Review of the literature and hypothesis authors should add some thesis regarding exactly the youth entrepreneurship.

RESPONSE: We thank you for your insightful remark. We agree with you and for that reason, we have modified the structure of the Introduction and the Review of the literature and hypothesis. We hope that the reader now has a more accurate view of what we want to convey.

5.) In the section: Materials and Methods the authors should explain why do they present the research results from 2013-2014 and why they chosen exactly this research method to verify the hypothesis.

RESPONSE: We are grateful for this suggestion that let us clarify and improve our manuscript. Our answer is two-fold:

Regarding the year 2013-2014, this is the year in which results were collected. This year was chosen because it allowed us a window to collect comparable information from students who followed the same educational curriculum, since it was already announced that the following academic year the curriculum would change, therefore making comparisons less reliable. As we mention in section 2.2. Participants and procedure “By collecting data in 2014, we were able to use more comparable data, as all participants were enrolled at school following the same curriculum (which was modified the following year by a new educational law in Spain, though with no effect on our data)”.

The applied neural network (CMLHL) was proposed as a novel method of performing Exploratory Projection Pursuit, in order to find more informative projections of data. We describe the justification for this research method in section 2.4. Data analysis: “As an exploratory technique, it can be applied when there is not target data to be reproduced by the neural network (as opposed to supervised learning). Hence, it fits the problem that the present paper addresses, in which there is a dataset that must be analyzed and whose internal structure has previously been unknown. To do so, different alternatives can be used and CMLHL is one of them that has previously been applied to a wide variety of problems. As it takes into account higher order statistics, its projections are more sparse and informative than those obtained by more traditional methods (such as Principal Component Analysis)”.

6.) In the section: Conclusions the authors should also present their recommendations for the bodies responsible for the educational policy as well as for some other entities responsible for the young entrepreneurs support such as: universities, incubators, commerce chambers, clusters and networks.

RESPONSE: Thank you for this remark. In this revised version of the manuscript, we have included recommendations for the bodies responsible for the educational policy as well as for other entities related to young entrepreneurs.

7.) Generally the issue is interesting but needs to be corrected at least according to the above remarks.

Good luck!

RESPONSE: Once again, we are extremely grateful for your insightful comments, which have been very valuable in improving the quality of our paper. We hope we have been able to address all your concerns adequately.

Reviewer 2 Report

I think the authors must include the references of Ajzen's works in order to better address the Theory of Planned Behaviour. In the same line, references to Liñan's works are needed when entrepreneurial intentions are mentioned, especially in the cases of works studying entrepreneurial education.

Author Response

Reviewer 2

I think the authors must include the references of Ajzen's works in order to better address the Theory of Planned Behaviour. In the same line, references to Liñan's works are needed when entrepreneurial intentions are mentioned, especially in the cases of works studying entrepreneurial education.

RESPONSE: Thank for pointing these issues to our attention, as the suggested works have helped us strengthen the theoretical framework of our paper. We have introduced references from the authors suggested by the reviewer, specifically:

Ajzen, I. The theory of planned behavior. Organ. Behav. Hum. Decis. Process. 1991, 50, 179–211. [CrossRef]

Ajzen, I. Nature and Operation of Attitudes. Ann. Rev. Psychol. 2001, 25, 27–58. [CrossRef] [PubMed]

Liñán, F.; Chen, Y.W. Development and cross-cultural application of a specific instrument to measure entrepreneurial intentions. Entrep. Theory Pract. 2009, 33, 593–617. [CrossRef]

Liñán, F. Skill and value perceptions: How do they affect entrepreneurial intentions? Int. Entrep. Manag. J. 2008, 4, 257–272. [CrossRef]

 Fayolle, A.; Liñán, F. The future of research on entrepreneurial intentions. J. Bus. Res. 2014, 67, 663–666. [CrossRef]